# Improving Stochastic Gradient Descent with Feedback

**Jayanth Koushik & Hiroaki Hayashi** *

Language Technologies Institute
Carnegie Mellon University
Pittsburgh, PA 15213, USA
{jkoushik,hiroakih}@cs.cmu.edu

## Abstract

In this paper we propose a simple and efficient method for improving stochastic gradient descent methods by using feedback from the objective function. The method tracks the relative changes in the objective function with a running average, and uses it to adaptively tune the learning rate in stochastic gradient descent. We specifically apply this idea to modify Adam, a popular algorithm for training deep neural networks. We conduct experiments to compare the resulting algorithm, which we call Eve, with state of the art methods used for training deep learning models. We train CNNs for image classification, and RNNs for language modeling and question answering. Our experiments show that Eve outperforms all other algorithms on these benchmark tasks. We also analyze the behavior of the feedback mechanism during the training process.

## 1 Introduction

Despite several breakthrough results in the last few years, the training of deep learning models remains a challenging problem. This training is a complex, high-dimensional, non-convex, stochastic optimization problem which is not amenable to many standard methods. Currently, the most common approach is to use some variant of stochastic gradient descent. Many extensions have been proposed to the basic gradient descent algorithm - designed to handle specific issues in the training of deep learning models. We review some of these methods in the next section.

Although variants of simple stochastic gradient descent work quite well in practice, there is still room for improvement. This is easily evidenced by the existence of numerous methods to simplify the optimization problem itself like weight initialization techniques and normalization methods.

In this work, we seek to improve stochastic gradient descent with a simple method that incorporates feedback from the objective function. The relative changes in the objective function indicate progress of the optimization algorithm. Our main hypothesis is that incorporating information about this change into the optimization algorithm can lead to improved performance - quantified in terms of the progress rate. We keep a running average of the relative changes in the objective function and use it to divide the learning rate. When the average relative change is high, the learning rate is reduced. This can improve the progress if, for example, the algorithm is bouncing around the walls of the objective function. Conversely, when the relative change is low, the learning rate is increased. This can help the algorithm accelerate through flat areas in the loss surface. As we discuss in the next section, such "plateaus" pose a significant challenge for first order methods and can create the illusion of local minima.

While our method is general i.e. independent of any particular optimization algorithm, in this work we specifically apply the method to modify Adam (Kingma & Ba, 2014), considered to be the state of the art for training deep learning models. We call the resulting algorithm Eve and design experiments to compare it with Adam, as well as other popular methods from the literature.

The paper is organized as follows. In Section 2, we review recent results related to the optimization of deep neural networks. We also discuss some popular algorithms and their motivations. Our general

---

*Equal contribution.

method, and the specific algorithm Eve are discussed in Section 3. Then in Section 4, we show that Eve consistently outperforms other methods in training convolutional neural networks (CNNs), and recurrent neural networks (RNNs). We also look in some detail, at the behavior of our method in the simple case of convex non-stochastic optimization. Finally we conclude in Section 5.

## 2 RELATED WORK

There has been considerable effort to understand the challenges in deep learning optimization. Intuitively, it seems that the non-convex optimization is made difficult by the presence of several poor local optima. However, this geometric intuition proves to be inadequate in reasoning about the high-dimensional case that arises with training deep learning models. Various empirical and theoretical results (Bray & Dean, 2007; Dauphin et al., 2014) have indicated that the problem in high dimensions arises not from local minima, but rather from *saddle points*. Moreover, a recent paper (Kawaguchi, 2016) *proved* (for deep linear networks, and under reasonable assumptions, also for deep non-linear networks) that all local minima in fact achieve the same value, and are optimal. The work also showed that all critical points which are not global minima are saddle points. Saddle points can seriously hamper the progress of both first and second order methods. Second order methods like Newton's method are actually attracted to saddle points and are not suitable for high dimensional non-convex optimization. First order methods can escape from saddle points by following directions of negative curvature. However, such saddle points are usually surrounded by regions of small curvature - plateaus. This makes first order methods very slow near saddle points and can create the illusion of a local minimum.

To tackle the saddle point problem, Dauphin et al. (2014) propose a second order method that fixes the issue with Newton's method. Their algorithm builds on considering the behavior of Newton's method near saddle points. Newton's method rescales gradients in each eigen-direction with the corresponding inverse eigenvalue. However, near a saddle point, negative eigenvalues can cause the method to move *towards* the saddle point. Based on this observation, the authors propose using the absolute values of the eigenvalues to rescale the gradients. This saddle-free Newton method is backed by theoretical justifications and empirical results; however due to the computational requirements, second order methods are not very suitable for training large scale models. So we do not compare with such approaches in this work.

We instead focus on first order methods which only rely on the gradient information. A key issue in training deep learning models is that of sparse gradients. To handle this, Adagrad (Duchi et al., 2011) adaptively changes the learning rate for each parameter, performing larger updates for infrequently updated parameters. However its update rule causes the learning rate to monotonically decrease, which eventually stalls the algorithm. Adadelta (Zeiler, 2012) and RMSProp (Tieleman & Hinton, 2012) are two extensions that try to fix this issue. Finally, a closely related method, and the base for our algorithm Eve (introduced in the next section), is Adam (Kingma & Ba, 2014). Adam incorporates the advantages of both Adagrad and RMSProp - and it has been found to work quite well in practice. Adam uses a running average of the gradient to determine the direction of descent, and scales the learning rate with a running average of the gradient squared. The authors of Adam also propose an extension based on the infinity norm, called Adamax. In our experiments, we compare Eve with both Adam and Adamax.

## 3 METHOD

### 3.1 ASSUMPTION

We do need to make an assumption about the objective function $f$. We assume that the minimum value of $f$ over its domain is known. While this is true for loss functions encountered in machine learning (like mean squared error or cross entropy), it does not hold if the objective function also includes regularization terms ($L_1$, $L_2$ etc.). In all our experiments, we used dropout for regularization which is not affected by this assumption. Finally, to simplify notation we assume that the minimum has been subtracted from the objective function i.e. the minimum has been made 0.

---

**Algorithm 1** Eve: Adam with feedback. Parameters carried over from Adam have the same default values: $\alpha = 10^{-3}$, $\beta_1 = 0.9$, $\beta_2 = 0.999$, $\epsilon = 10^{-8}$. For parameters specific to our method, we recommend default values $\beta_3 = 0.999$, $k = 0.1$, $K = 10$. Wherever applicable, products are elementwise.

---

**Require:** $\alpha$: learning rate
**Require:** $\beta_1, \beta_2 \in [0, 1)$: exponential decay rates for moment estimation in Adam
**Require:** $\beta_3 \in [0, 1)$: exponential decay rate for computing relative change
**Require:** $k, K$: lower and upper threshold for relative change
**Require:** $\epsilon$: fuzz factor
**Require:** $f(\theta)$: objective function
**Require:** $\theta_0$: initial value for parameters

$\quad m_0 = v_0 = 0$
$\quad d_0 = 1$
$\quad \widehat{f}_{-1} = t = 0$
$\quad$**while** stopping condition is not reached **do**
$\quad\quad t \leftarrow t + 1$
$\quad\quad g_t \leftarrow \nabla_\theta f(\theta_{t-1})$
$\quad\quad m_t \leftarrow \beta_1 m_{t-1} + (1 - \beta_1)g_t$
$\quad\quad \widehat{m}_t \leftarrow \frac{m_t}{(1 - \beta_1^t)}$
$\quad\quad v_t \leftarrow \beta_2 v_{t-1} + (1 - \beta_2)g_t^2$
$\quad\quad \widehat{v}_t \leftarrow \frac{v_t}{(1 - \beta_2^t)}$
$\quad\quad$**if** $t > 1$ **then**
$\quad\quad\quad$**if** $f(\theta_{t-1}) \geq \widehat{f}_{t-2}$ **then**
$\quad\quad\quad\quad \delta_t \leftarrow k + 1$
$\quad\quad\quad\quad \Delta_t \leftarrow K + 1$
$\quad\quad\quad$**else**
$\quad\quad\quad\quad \delta_t \leftarrow \frac{1}{K+1}$
$\quad\quad\quad\quad \Delta_t \leftarrow \frac{1}{k+1}$
$\quad\quad\quad$**end if**
$\quad\quad\quad c_t \leftarrow \min\left\{\max\left\{\delta_t, \frac{f(\theta_{t-1})}{\widehat{f}_{t-2}}\right\}, \Delta_t\right\}$
$\quad\quad\quad \widehat{f}_{t-1} \leftarrow c_t \widehat{f}_{t-2}$
$\quad\quad\quad r_t \leftarrow \frac{|\widehat{f}_{t-1} - \widehat{f}_{t-2}|}{\min\{\widehat{f}_{t-1}, \widehat{f}_{t-2}\}}$
$\quad\quad\quad d_t \leftarrow \beta_3 d_{t-1} + (1 - \beta_3)r_t$
$\quad\quad$**else**
$\quad\quad\quad \widehat{f}_{t-1} \leftarrow f(\theta_{t-1})$
$\quad\quad\quad d_t \leftarrow 1$
$\quad\quad$**end if**
$\quad\quad \theta_t \leftarrow \theta_{t-1} - \alpha \frac{\widehat{m}_t}{d_t\sqrt{\widehat{v}_t} + \epsilon}$
$\quad$**end while**
$\quad$**return** $\theta_t$

---

## 3.2 ALGORITHM

The main component of our proposed method is a feedback term that captures the relative change in the objective value. Let $f_{t-2}$ and $f_{t-1}$ denote the values of the objective function at time steps $t - 2$ and $t - 1$ respectively. Then this change is computed as $r_t = \frac{f_{t-2} - f_{t-1}}{f_{t-1}} = \frac{f_{t-2} - f_{t-1}}{f_{t-1}}$ if $f_{t-2} \geq f_{t-1}$, and $\frac{f_{t-1} - f_{t-2}}{f_{t-2}}$ otherwise. Note that this value is always non-negative but it can be less than or greater than 1 i.e. it captures both relative increase and decrease. We compute a running average using these relative changes to get a smoother estimate. Specifically, we take $d_1 = 1$, and for $t > 1$ define $d_t$ as $d_t = \beta d_{t-1} + (1 - \beta)r_t$. Here $\beta \in [0, 1)$ is a decay rate - large values correspond to a slowly changing $d_t$, and vice versa. This simple expression can, however, blow up and lead to instability. To handle this issue, we use a thresholding scheme. A simple thing to do would be to clip $d_t$ as $\min\{\max\{k, d_t\}, K\}$ for some suitable $0 < k < K$. But we found this to not work very well in practice due to the abrupt nature of the clipping. Instead we indirectly clip $d_t$ by smoothly tracking

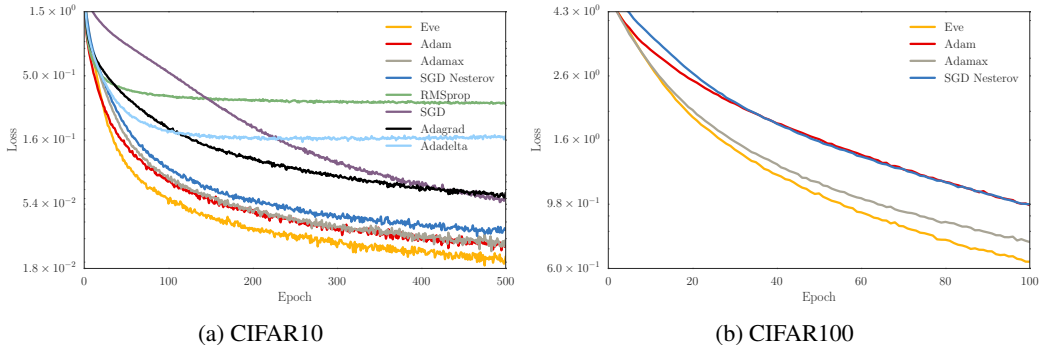

(a) CIFAR10 (b) CIFAR100

Figure 1: Training loss for convolutional neural networks on CIFAR10 and CIFAR100. The vertical axis is shown in log scale. In both cases, our proposed method achieves the best performance.

the objective function. Let $\widehat{f}_{t-2}$ be the value of the smoothly tracked objective function at time $t-2$ with $\widehat{f}_0 = f_0$. For now, assume $f_{t-1} \leq \widehat{f}_{t-2}$. We would like to have $k \leq r_t \leq K$ which in this case requires $\frac{f_{t-1}}{\widehat{f}_{t-2}} \in \left[\frac{1}{K+1}, \frac{1}{k+1}\right]$. So we compute $c_t = \min\left\{\max\left\{\frac{1}{K+1}, \frac{f_t}{\widehat{f}_{t-1}}\right\}, \frac{1}{k+1}\right\}$ and set $\widehat{f}_{t-1} = c_t \widehat{f}_{t-2}$. Finally $r_t$ is $\frac{\widehat{f}_{t-2} - \widehat{f}_{t-1}}{\widehat{f}_{t-1}}$. Analogous expressions can also be derived for the case when $f_{t-1} > \widehat{f}_{t-2}$. This smooth tracking has the additional advantage of making $d_t$ less susceptible to the high variability that comes with training using minibatches.

Once $d_t$ has been computed, it can be used to modify any gradient descent algorithm by modifying the learning rate $\alpha$ as $\alpha_t = \alpha/d_t$. Large values of $d_t$, caused by large changes in the objective function will lead to a smaller effective learning rate. Similarly, small values of $d_t$ will lead to a larger effective learning rate. Since we start with $d_0 = 1$, the initial updates will closely follow that of the base algorithm. In the next section, we will look at how $d_t$ evolves during the course of an experiment to get a better understanding of how it affects the training.

We note again that our method is independent of any particular gradient descent algorithm. However, for this current work, we specifically focus on applying the method to Adam (Kingma & Ba, 2014). This modified algorithm, which we call Eve, is shown in Algorithm 1. We modify the final Adam update by multiplying the denominator $\sqrt{v_t}$ with $d_t$. In addition to the hyperparameters in Adam, we introduce 3 new hyperparameters $\beta_3$, $k$, and $K$. In all our experiments we use the values $\beta_3 = 0.999$, $k = 0.1$, and $K = 10$, which we found to work well in practice.

## 4 EXPERIMENTS

Now we evaluate our proposed method by comparing Eve with several state of the art algorithms for optimizing deep learning models.[1]. In all experiments, we used ReLU activations, and initialized weights according to the scheme proposed by Glorot & Bengio (2010). We used minibatches of size 128, and linear decay for the learning rate: $\alpha_t = \alpha/(1 + \gamma t)$ ($\gamma$ is the decay rate, picked by searching over a range of values).

In the figures, SGD refers to vanilla stochastic gradient descent, and SGD Nesterov refers to stochastic gradient descent with Nesterov momentum (Nesterov, 1983) where we set the momentum to 0.9 in all experiments.

### 4.1 CONVOLUTIONAL NEURAL NETWORKS

We first trained a 5 layer convolutional neural network for 10-way classification of images from the CIFAR10 dataset (Krizhevsky & Hinton, 2009). The model consisted of 2 blocks of 3x3 convolutional layers each followed by 2x2 max pooling and 0.25 dropout (Srivastava et al., 2014). The first block

---

[1]Full technical details of the experiments and additional results are available at https://github.com/jayanthkoushik/sgd-feedback.

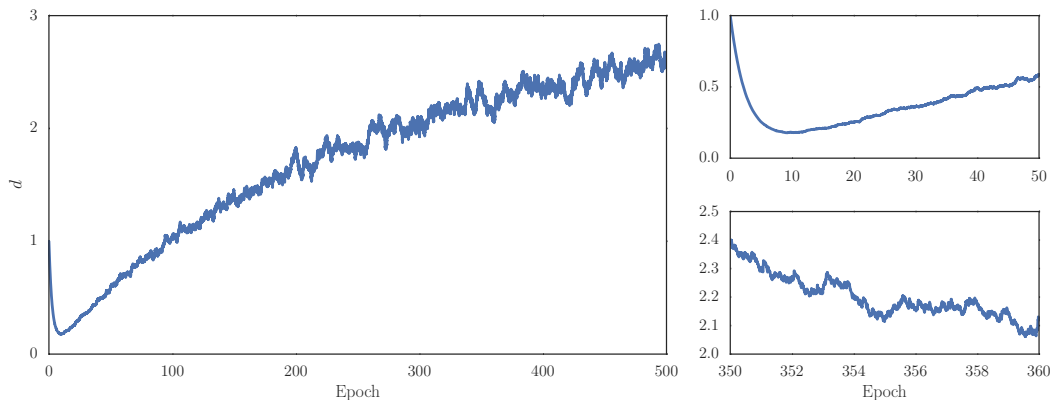

Figure 2: Behavior of the tuning coefficient $d_t$ during the experiment shown in Figure 1a. There is an overall trend of acceleration followed by decay, but also more fine-grained behavior as indicated by the bottom-right plot.

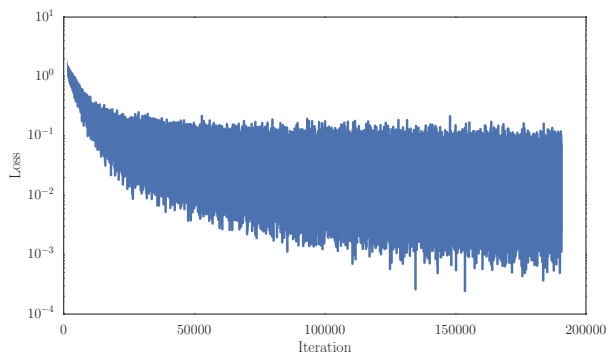

Figure 3: Minibatch losses for Eve during the CNN experiment on CIFAR10 (Figure 1a). The variance in the losses increases throughout the training.

contained 2 layers with 32 filters each, and the second block contained 2 layers with 64 filters each. The convolutional layers were followed by a fully connected layer with 512 units and a 10-way softmax layer. We trained this model for 500 epochs on the training split using various popular methods for training deep learning models, as well as Eve. For each algorithm, we tried learning rates $\{10^{-2}, 10^{-3}, 10^{-4}\}$ (for algorithms with suggested default learning rates, we also included them in the search), learning rate decays $\{0, 10^{-2}, 10^{-3}, 10^{-4}\}$, and picked the pair of values that led to the smallest final training loss. The loss curves are shown in Figure 1a. Eve quickly surpasses all other methods and achieves the lowest final training loss. In the next section we will look at the behavior of the adaptive coefficient $d_t$ to gain some intuition behind this improved performance.

We also trained a larger CNN model using the top-performing algorithms from the previous experiment. This model consisted of 3 blocks of 3x3 convolutional layers (3 layers per block, and 64, 128, 256 filters per layer in the first, second, and third block respectively) each followed by 2x2 max pooling and 0.5 dropout. Then we had 2 fully connected layers with 512 units, each followed by 0.5 dropout, and finally a 100-way softmax. We trained this model on the CIFAR100 (Krizhevsky & Hinton, 2009) dataset for 100 epochs. We again performed a grid search over the same learning rate and decay values as the last experiment. The results are shown in Figure 1b, and once again show that our proposed method improves over state of the art methods for training convolutional neural networks.

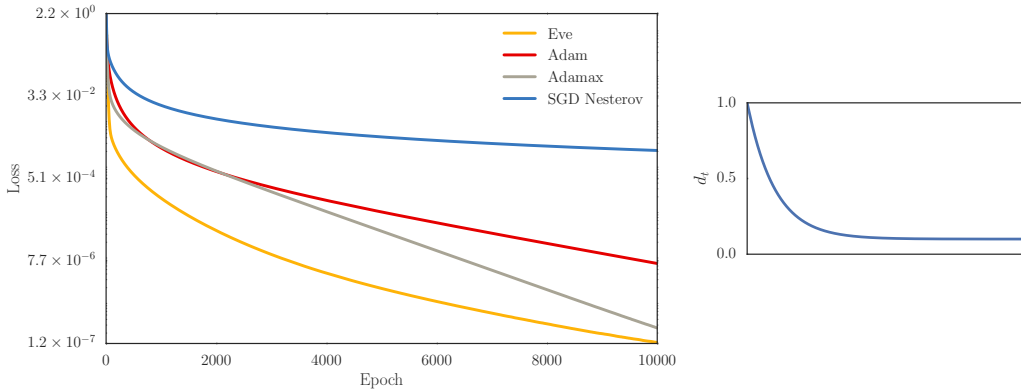

Figure 4: Loss curves and tuning coefficient $d_t$ for batch gradient descent training of a logistic regression model. For Eve, $d_t$ continuously decreases and converges to the lower threshold 0.1.

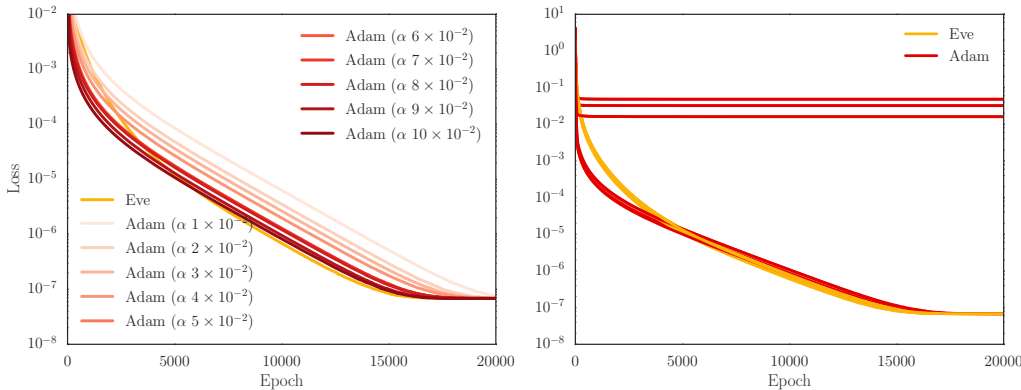

Figure 5: The left plot shows Adam with different learning rates plotted with Eve (learning rate $10^{-2}$). Although Adam with a larger learning rate can be almost identical to Eve, this is largely dependent on the initial values for Adam as shown in the plot on the right.

## 4.2 ANALYSIS OF TUNING COEFFICIENT

Before we consider the next set of experiments on recurrent neural networks, we will first look more closely at the behavior of the tuning coefficient $d_t$ in our algorithm. We will specifically consider the results from the CNN experiment on CIFAR10. Figure 2 shows the progress of $d_t$ throughout the training, and also in two smaller windows. A few things are worth noting here. First is that of the overall trend. There is an initial acceleration followed by a decay. This initial acceleration allows Eve to rapidly overtake other methods, and makes it proceed at a faster pace for about 100 epochs. This acceleration is not equivalent to simply starting with a larger learning rate - in all our experiments we search over a range of learning rate values. The overall trend for $d_t$ can be explained by looking at the minibatch losses at each iteration (as opposed to the loss computed over the entire dataset after each epoch) in Figure 3. Initially, different minibatches achieve similar loss values which leads to $d_t$ decreasing. But as training proceeds, the variance in the minibatch losses increases and $d_t$ eventually increases. However, this overall trend does not capture the complete picture - for example, as shown in the bottom right plot of Figure 2, $d_t$ can actually be decreasing in some regions of the training, adjusting to local structures in the error surface.

To further study the observed acceleration, and to also motivate the need for clipping, we consider a simpler experiment. We trained a logistic regression model on 1000 images from the MNIST dataset. We used batch gradient descent for training i.e. all 1000 samples were used for computing the gradient at each step. We trained this model using Eve, Adam, Adamax, and SGD Nesterov

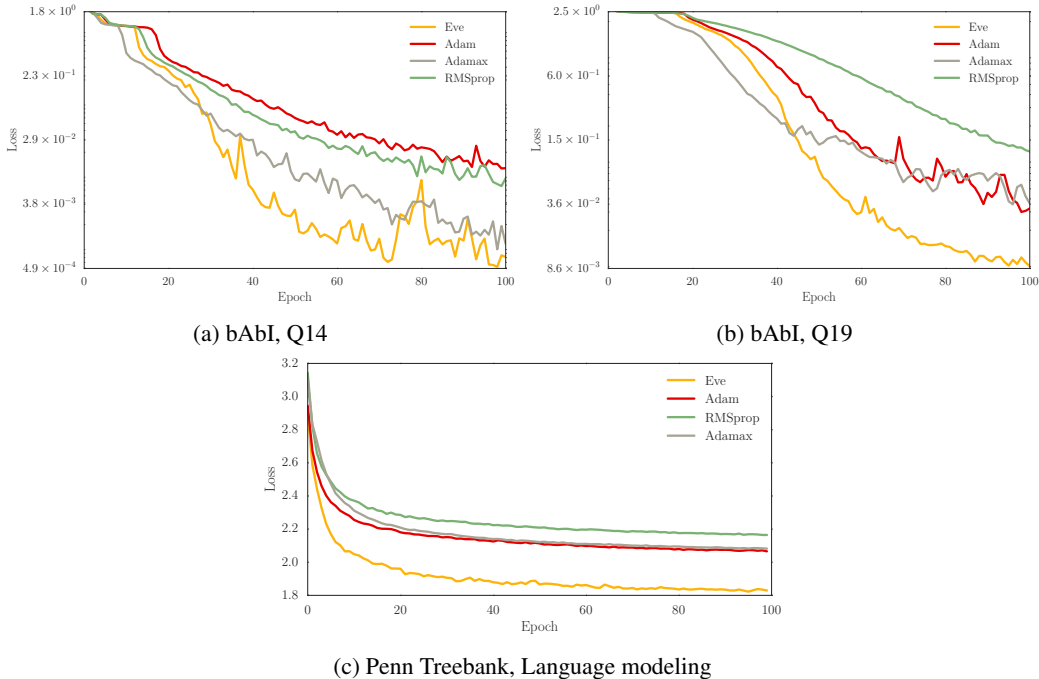

(a) bAbI, Q14

(b) bAbI, Q19

(c) Penn Treebank, Language modeling

Figure 6: Loss curves for experiments with recurrent neural networks. Eve consistently achieves better performance than other methods.

for 10000 iterations, searching over a large range of values for the learning rate and decay: $\alpha \in \{10^{-1}, 10^{-2}, 10^{-3}, 10^{-4}, 10^{-5}\}$, $\gamma \in \{10^{-2}, 10^{-3}, 10^{-4}, 10^{-5}, 10^{-6}, 0\}$. The results are shown in Figure 4. Eve again outperforms all other methods and achieves the lowest training loss. Also, since this is a smooth non-stochastic problem, the tuning coefficient $d_t$ continuously decreases - this makes having a thresholding mechanism important since the learning rate would blow up otherwise.

Although in the previous experiment the effect of our method is to increase the learning rate, it is not equivalent to simply starting with a larger learning rate. We will establish this with a couple simple experiments. First we note that in the previous experiment, the optimal decay rates for both Adam and Eve were 0 - no decay. The optimal learning rate for Eve was $10^{-2}$. Since the tuning coefficient converges to 0.1, we trained Adam using no decay, and learning rates $i \times 10^{-2}$ where $i$ varies from 1 to 10. The training loss curves are shown in the left plot of Figure 5. While increasing the learning rate with Adam does seem to close the gap with Eve, Eve does remain marginally faster. Moreover, and more importantly, this first plot represents the best-case situation for Adam. With larger learning rates, training becomes increasingly unstable and sensitive to the initial values of the parameters. This is illustrated in the right plot of Figure 5 where we used Eve (with learning rate $10^{-2}$) and Adam (with learning rate $10^{-1}$) 10 times with different random initializations. In some cases, Adam fails to converge whereas Eve *always* converges - even though Eve eventually reaches a learning rate of $10^{-1}$. This is because very early in the training, the model is quite sensitive at higher learning rates due to larger gradients. Depending on the initial values, the algorithm may or may not converge. So it is advantageous to slowly accelerate as the learning stabilizes rather than start with a larger learning rate.

## 4.3 RECURRENT NEURAL NETWORKS

Finally, we evaluated our method on recurrent neural networks (RNNs). We first trained a RNN for character-level language modeling on the Penn Treebank dataset (Marcus et al., 1993). Specifically, the model consisted of a 2-layer character-level Gated Recurrent Unit (Chung et al., 2014) with hidden layers of size 256, 0.5 dropout between layers, and sequences of 100 characters. We adopted $10^{-3}$ as the initial learning rate for Adam, Eve, and RMSProp. For Adamax, we used $2 \times 10^{-3}$ as the learning rate since it is the suggested value. We used $3 \times 10^{-4}$ for the learning rate decay. We

trained this model for 100 epochs using each of the algorithms. The results, plotted in Figure 6c, clearly show that our method achieves the best results. Eve optimizes the model to a lower final loss than the other models.

We trained another RNN-based model for the question & answering task. Specifically, we chose two question types among 20 types from the bAbI dataset (Weston et al., 2015), Q19 and Q14. The dataset consists of pairs of supporting story sentences and a question. Different types of pairs are said to require different reasoning schemes. For our test case, Q19 and Q14 correspond to *Path Finding* and *Time Reasoning* respectively. We picked Q19 since it is reported to have the lowest baseline score, and we picked Q14 randomly from the remaining questions. The model consisted of two parts, one for encoding story sentences and another for query. Both included an embedding layer with 256 hidden units, and 0.3 dropout. Next query word embeddings were fed into a GRU one token at a time, to compute a sentence representation. Both story and query sequences were truncated to the maximum sequence length of 100. Finally, the sequence of word embeddings from story sentences and the repeated encoded representation of a query were combined together to serve as input for each time step in another GRU, with 0.3 dropout. We searched for the learning rate and decay from a range of values, $\alpha \in \{10^{-2}, 10^{-3}, 10^{-4}\}$ and $\gamma \in \{10^{-2}, 10^{-3}, 10^{-4}, 0\}$. The results, shown in Figures 6a, and 6b show that Eve again improves over all other methods.

## 5 CONCLUSION

We proposed a simple and efficient method for incorporating feedback in to stochastic gradient descent algorithms. We used this method to create Eve, a modified version of the Adam algorithm. Experiments with a variety of models showed that the proposed method can help improve the optimization of deep learning models.

For future work, we would look to theoretically analyze our method and its effects. While we have tried to evaluate our algorithm Eve on a variety of tasks, additional experiments on larger scale problems would further highlight the strength of our approach. We are making code for our method and the experiments publicly available to encourage more research on this method.

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
