# Peer review of "Improving Stochastic Gradient Descent with Feedback"

_ICLR 2017 — rejected_

[Public Comment · Timo Aila · 22 Nov 2016]
**Error in pseudocode**

I think there is a bug in Algorithm 1. The comparison between f and \hat{f} should be the other way around.

With that fix, I re-implemented the algorithm and indeed it was slightly faster than Adam in training a complex autoencoder :)

[Official Review · AnonReviewer1 · rating 6 · confidence 4 · 16 Dec 2016]
**A learning rate tuning method for Adam**

The paper demonstrates a semi-automatic learning rate schedule for the Adam optimizer, called Eve. Originality is somehow limited but the method appears to have a positive effect on neural network training. The paper is well written and illustrations are appropriate.

Pros:

- probably a more sophisticated scheduling technique than a simple decay term
- reasonable results on the CIFAR dataset (although with comparably small neural network)

Cons:

- effect of momentum term would be of interest
- the Adam reference doesn't point to the conference publications but only to arxiv
- comparison to Adam not entirely conclusive

[Official Review · AnonReviewer2 · rating 5 · confidence 4 · 16 Dec 2016]
**d_t**

As you noted for Figure 5 Left, sometimes it seems sufficient to tune learning rates. I see your argument for Figure 6 Right, 
but 
1) not for all good learning rates make Adam fail, I guess you selected the one where it did (note that Adam was several times faster than Eve in the beginning)
2) I don't buy "Eve always converges" because you show it only for 0.1 and since Eve is not Adam, 0.1 of Adam is not 0.1 of Eve because of d_t. 

To my understanding, you define d_t over time with 3 hyperparameters. Similarly, one can define d_t directly. The behaviour of d_t that you show is not extraordinary and can be parameterized. If Eve is better than Adam, then looking at d_t we can directly see whether we underestimated or overestimated learning rates. You could argue that Eve does it automatically but you do tune learning rates for each problem individually anyway.

[Official Review · AnonReviewer3 · rating 5 · confidence 4 · 20 Dec 2016]

The paper introduced an extension of Adam optimizer that automatically adjust learning rate by comparing the subsequent values of the cost function during training. The authors empirically demonstrated the benefit of the Eve optimizer on CIFAR convnets, logistic regression and RNN problems.

I have the following concerns about the paper

- The proposed method is VARIANT to arbitrary shifts and scaling to the cost function.  

- A more fair comparison with other baseline methods would be using additional exponential decay learning scheduling between the lower and upper threshold of d_t. I suspect 1/d_t just shrinks as an exponential decay from Figure 2.

- Three additional hyper-parameters: k, K, \beta_3.

Overall, I think the method has its fundamental flew and the paper offers very limited novelty. There is no theoretical justification on the modification, and it would be good for the authors to discuss the potential failure mode of the proposed method. Furthermore, it is hard for me to follow Section 3.2. The writing quality and clarity of the method section can be further improved.

[Final Decision · Program Chairs · 06 Feb 2017]
**ICLR committee final decision**

The authors propose a simple strategy that uses function values to improve the performance of Adam. There is no theoretical analysis of this variant, but there is an extensive empirical evaluation. A disadvantage of the proposed approach is that it has 3 parameters to tune, but the same parameters are used across experiments. Overall however, the PCs believe that this paper doesn't quite reach the level expected for ICLR and thus cannot be accepted.